# Monocular Event-Based Vision for Obstacle Avoidance with a Quadrotor

**Anish Bhattacharya[1], Marco Cannici[2], Nishanth Rao[1], Yuezhan Tao[1],**
**Vijay Kumar[1], Nikolai Matni[1], Davide Scaramuzza[2]**

[1]GRASP, University of Pennsylvania
[2]Robotics and Perception Group, University of Zurich
https://www.anishbhattacharya.com/research/evfly

**Abstract:** We present the first static-obstacle avoidance method for quadrotors using just an onboard, monocular event camera. Quadrotors are capable of fast and agile flight in cluttered environments when piloted manually, but vision-based autonomous flight in unknown environments is difficult in part due to the sensor limitations of traditional onboard cameras. Event cameras, however, promise nearly zero motion blur and high dynamic range, but produce a very large volume of events under significant ego-motion and further lack a continuous-time sensor model in simulation, making direct sim-to-real transfer not possible. By leveraging depth prediction as a pretext task in our learning framework, we can pre-train a reactive obstacle avoidance events-to-control policy with approximated, simulated events and then fine-tune the perception component with limited events-and-depth real-world data to achieve obstacle avoidance in indoor and outdoor settings. We demonstrate this across two quadrotor-event camera platforms in multiple settings and find, contrary to traditional vision-based works, that low speeds (1m/s) make the task harder and more prone to collisions, while high speeds (5m/s) result in better event-based depth estimation and avoidance. We also find that success rates in outdoor scenes can be significantly higher than in certain indoor scenes.

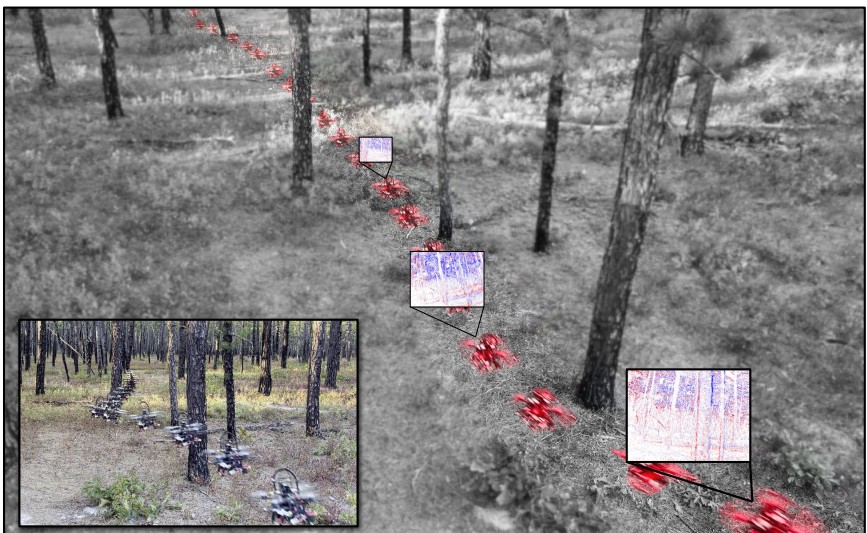

Figure 1: Our event camera-equipped quadrotor avoids static obstacles under considerable ego-motion. Our simulation pre-trained, events-to-control policy is fine-tuned with real-world perception data, such as that from a forest. Red lines indicate avoidance maneuvers during the flight. We demonstrate obstacle avoidance in indoor, outdoor, and dark environments.

8th Conference on Robot Learning (CoRL 2024), Munich, Germany.

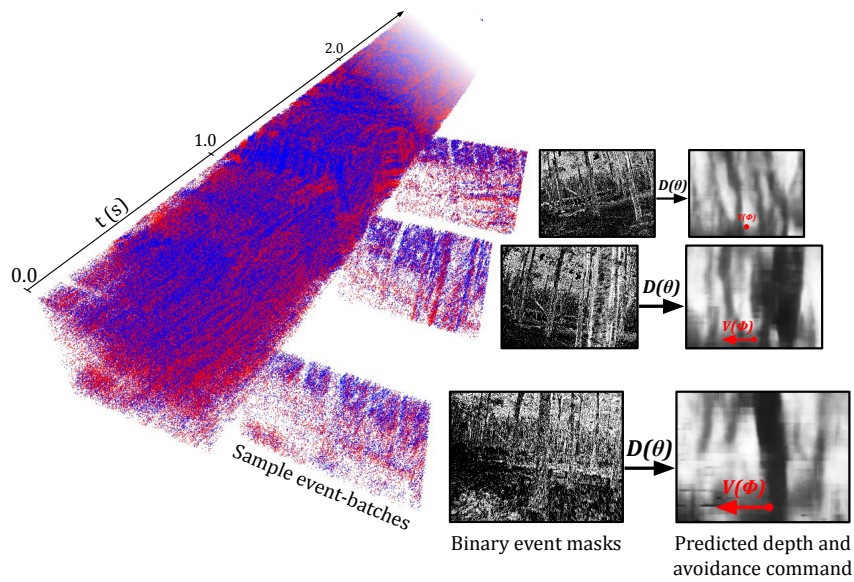

Figure 2: A portion of the continuous event stream is shown from the forest trial in Figure 1 (showing 5% of the true event stream density for viewability), with sample 33ms-batches and the corresponding model predictions.

# 1   Introduction

Quadrotors can be flown very fast and with high agility by human pilots. More recently, autonomous, vision-based quadrotors demonstrated champion-level drone racing on fixed tracks [1]. However, achieving high-speed vision-based autonomous flight through unknown environments remains challenging due to the inherent limitations of traditional cameras. Their limited frame rate, low dynamic range, and motion blur, particularly in low-lit scenes such as under forest canopy, can increase uncertainty in traditional model-based mapping-planning-control pipelines or can cause out-of-distribution inputs for an end-to-end vision-to-control learning framework.

In contrast, event cameras (also called dynamic vision sensors or neuromorphic cameras) [2] have low latency and low bandwidth and feature pixels that individually detect changes at microsecond-level temporal resolution, effectively alleviating motion blur. Their output event stream consists of positive or negative events that correspond to local, per-pixel increases or decreases in brightness. Therefore, unlike traditional cameras, which output frame-based data at constant time intervals, event cameras generate a continuous stream of events.

This rapid and adaptive perception capability, particularly under variable lighting conditions, suggests that an events-driven approach could enable vision-based agile flight at significantly faster speeds than traditional cameras. Most related works using event cameras have focused on dodging dynamic objects thrown at a flying quadrotor [3, 4, 5]. These works focused on detecting the dynamic object by eliminating events caused by the ego-motion of the robot with respect to the stationary background. To the best of our knowledge, there is no published work for *static*-obstacle avoidance due to the challenge of significant event-stream bandwidth under considerable ego-motion, which makes the obstacle identification task in the event stream more difficult.

We present a simulation pre-trained, learning-based approach to predict static-obstacle avoidance velocity commands from an incoming event stream, by leveraging depth prediction as a pretext task to guide the network in extracting robust, texture-invariant latent representations. Focusing on geometry rather than appearance, depth offers a more domain-invariant representation and, for these reasons, has been employed in various robotics tasks. It can easily be gathered in the real world for model fine-tuning, offering an inexpensive alternative compared to expert drone pilot commands.

We focus on tree-like objects and demonstrate that jointly pre-training velocity and depth prediction in simulation allows us to fine-tune the perception backbone with real-world data, leading to the successful avoidance of trees in a forest. We tackle the lack of a continuous-time event camera model in simulation by using a binarized brightness difference event representation, which is less sensitive to non-continuous event streams, and employ a teacher-student approach to distill privileged knowledge into a vision-only student network. We demonstrate simulated, real indoor, and real outdoor avoidance of trees and tree-like objects with a single event camera onboard a flying quadrotor, deployed on two different hardware platforms, showcasing the adaptability of the proposed system.

**Our contributions are as follows:**

- The first events-driven static-obstacle avoidance method for a mobile robot.
- Successful sim-to-real, few-shot transfer of an events-based policy by leveraging depth prediction as a pretext, supervisable task.
- High-speed (5m/s) real-world avoidance of trees with purely onboard computation, and demonstrating that avoidance using event-based vision improves as speed increases.
- Open-source code and data for simulation, data collection, training, and testing.

## 2 Related Work

**Vision-based obstacle avoidance with drones.** Various approaches have been developed to address vision-based obstacle avoidance with quadrotors in cluttered and complex environments. Previous literature has explored reinforcement learning with high-fidelity renderers and robust latent representations [6, 7, 1], unsupervised learning strategies interacting directly with the environment [8] or using weakly supervised data [9, 10], as well as supervised methods using annotated data [11] and expert policy distillation [12]. While neural network-based policies directly regressing action commands from images [9, 13, 11] have been proposed, depth has recently emerged as a more robust modality for deployment, providing essential obstacle information while discarding unnecessary texture details. Leveraging depth as a primary sensory input, Loquercio et al. [12] demonstrate zero-shot transfer capabilities in the real world while only training in simulation using a student-teacher approach. A similar approach is used by Bhattacharya et al. [14], who designed transformer- and recurrent-based policies that directly regress control commands from depth. Inspired by these works, we propose to also exploit depth representation for training; however, since we only use events as policy input, we propose to use depth as an auxiliary task to learn a robust latent representation and promote the learning of texture-invariant features for optimal real-world transfer.

**Control from event-based vision.** Event cameras have motivated a rapidly growing body of work in vision-based robotics, specifically in learning reactive policies that directly map sparse event data into control commands. Fast, reactive policies have been demonstrated in the task of catching fast-moving objects with quadruped robots using reinforcement learning [15], and in drones using model-based algorithms to avoid fast dynamic obstacles from hovering conditions [4]. In the domain of drone flight, a neuromorphic pipeline was first demonstrated by Vitale et al. [16], using a spiking neural network (SNN) for line-tracking with a constrained 1-DoF dualcopter. Similarly, Paredes-Vallés et al. [17] employed a neuromorphic vision-to-control approach for real-time ego-motion estimation and control of an unconstrained drone, achieving successful sim-to-real transfer for various flight tasks. Additionally, event-based object detection has been explored as an intermediate task for obstacle avoidance and navigation. Zhang et al. utilized an SNN combined with a depth camera for object detection [18], while Andersen et al. [19] trained a convolutional and recurrent network to detect gates under fast motion in drone racing. Similar to these works, we propose to learn a vision-to-control policy with event cameras; however, we follow an approach inspired by that proposed in Loquercio et al. [12], and use a teacher-student training approach where a model-based teacher with privileged information is distilled in a vision-based student network in simulation. By using depth estimation as a pretext task and limited real-world annotated data for fine-tuning, we

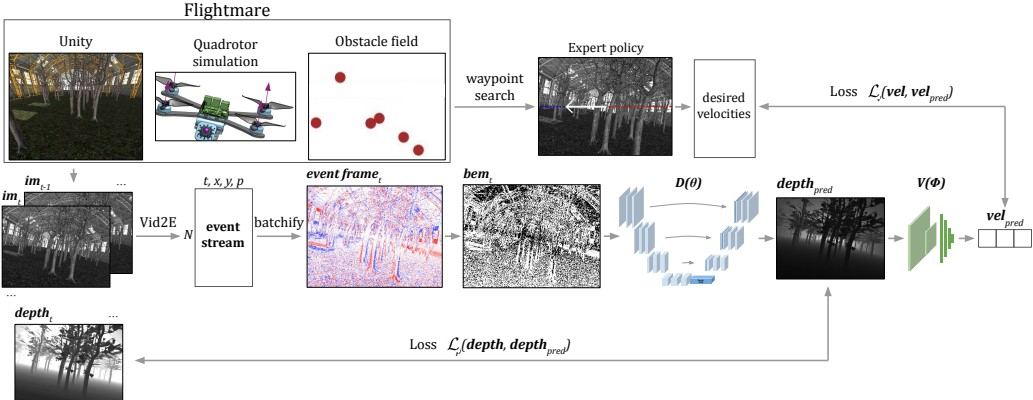

Figure 3: Data generation and learning framework. Grayscale images and depths are collected from quadrotor flights in Flightmare [20]. Vid2E [21] processes the images to generate an event stream, which is batched and converted to binary event masks (BEMs) representations for input to the depth predictor $D(\theta)$. The perception loss $\mathcal{L}_p$ is computed relative to the ground truth depth. Ground truth obstacle states inform the expert policy, which logs desired velocities for supervising the velocity predictor $V(\phi)$. Multi-dimensional quantities are in bold.

demonstrate effective sim-to-real transfer, enabling reliable drone flight and obstacle avoidance in multiple real-world scenarios.

## 3 Methodology

Our approach addresses the challenge posed by the lack of an effective continuous-time event camera simulator despite the need to transfer our model to continuous-time event streams in the real world.

### 3.1 Simulation setup and data gathering

We utilize the Flightmare simulator [20] to run a privileged expert policy on a flying quadrotor while collecting perception (grayscale and depth images) and command (velocity) data (Figure 3). As there is no native, real-time event simulator available, we leverage the photorealistic and high-fidelity images from Unity to generate event-based data (Section 3.2). We use a modified version of the privileged expert policy described in Bhattacharya et al. [14], which uses receding-horizon planning to execute velocity commands towards free-space waypoints. We alter the waypoint search to be along a horizon-aligned direction for avoiding trees (see Figure 3, *Expert policy*). The student velocity predictor $V(\phi)$ predicts a scalar velocity magnitude $v_y \in [-1, 1]$ in the lateral direction, and since it is assumed $v_z = 0$, the forward velocity command component $v_x$ is computed via $||v_x + v_y||_2^2 = 1$. This vector is then scaled up to a velocity command based on a desired speed.

### 3.2 Event data generation and representation

A single event is represented as $e_k = (t_k, x_k, y_k, p_k)$, denoting a brightness change registered by an event camera at time $t_k$, pixel location $(x_k, y_k)$ in the event pixel array, with polarity $p_k \in \{-1, +1\}$. The polarity of an event indicates a positive or negative change in logarithmic illumination, quantized by positive and negative thresholds $C^{\pm}$.

We use Vid2E [21] to process grayscale images and generate events. While Vid2E provides an option of up-sampling the input video via deep learning-based methods (such as Super SloMo [22]), our learning framework requires hundreds of trajectories of data for training, such that the adaptive up-sampling is impossible due to both long processing time and disk storage limits. Therefore, we must use Vid2E without up-sampling, which results in a more discontinuous event stream. This

necessitates the batching of simulation-generated events into time windows discretized by the framerate of the camera, minimizing the harmful impact of the discontinuities in selecting batches. This eliminates the possibility of using a voxel grid representation of events [23], which may take better advantage of the high temporal resolution of the neuromorphic sensor.

The number of positive and negative events at each pixel location is summed according to their polarity, such that equal numbers of positive and negative events at a pixel within the time window $\Delta t = 1/\text{FPS}_{cam}$ results in a zero count. Edges of nearby objects are accentuated with this way of counting events (generally with leading/trailing edges of opposite polarities for passing objects, and the same polarity for head-on objects) relative to the object's interior or other ground or complex textures. We then convert this to a binary event mask (BEM), which we found resulted in improved training over the histogram array of event counts. Given event stream $E$, an element of the BEM at a given pixel location $(x, y)$ is as follows.

$$\mathbf{BEM}_{x,y} = \mathbb{1}\left(\left(\sum_{e \in E} \mathbb{1}(e_p = +1) - \sum_{e \in E} \mathbb{1}(e_p = -1)\right) \neq 0\right) \tag{1}$$

### 3.3 Learning framework

The perception backbone $D(\theta)$ of the learning framework consists of a U-Net architecture [24] to predict depth from the input BEM, inspired by works that convert between images and events [21] and predict dense depth from monocular events in particular settings [25]. The U-Net model is popular for segmentation tasks in medical imaging and modern computer vision works; this lends well to obstacle avoidance since obstacles must be segmented from the background. A formulation close to the original U-Net was used, except that interpolations were used for skip connections rather than crops, and a ConvLSTM [26] was used after the middle layers, prior to the transpose convolutions, to promote temporal consistency in generated depth images over time. Depth images gathered through expert rollouts provided supervision of this module via an L2 perception loss $\mathcal{L}_p$, weighted at each pixel by the inverse ground truth depth value at that pixel, thereby prioritizing nearby objects in each gradient step.

The velocity predictor $V(\phi)$ takes as input the depth prediction following the transposed convolutions and consists of a convolutional neural network ending in fully connected layers, producing the single scalar avoidance velocity $v_y$. This velocity predictor is trained jointly with the perception backbone and also utilizes an L2 loss function $\mathcal{L}_v$ with the expert's commanded velocity. The weights used on the perception $\mathcal{L}_p$ and velocity $\mathcal{L}_v$ loss terms are $(1, 10)$.

$$\mathcal{L}_{p_{i,j}} = \frac{1}{\mathbf{depth}_{i,j}} \left\|\mathbf{depth}_{i,j} - \widehat{\mathbf{depth}}_{i,j}\right\|_2^2 \quad , \qquad \mathcal{L}_v = \|v_y, \widehat{v_y}\|_2^2 \tag{2}$$

### 3.4 Few-shot transfer to the real world and to other platforms

Without an expert pilot, we can only gather real-world perception data for fine-tuning the perception backbone. For a given scene, we collect aligned event batch and depth image pairs with a hand-held rig of an event camera rigidly mounted with an Intel Realsense D435 depth camera, calibrated via E2Calib [27] (see supplementary). While simulation datasets ranged from 100-200 long trajectories, real datasets used for fine-tuning consisted of 10-20 shorter trajectories. Frame-batching of events enables image-based data augmentation techniques that particularly apply to the events-based quadrotor fly-and-avoid task, such as applying small rotations (simulating roll angles), left-right flipping, and adding noise. Applying these augmentations to real datasets drastically improved performance during deployment. See the supplementary for details on the impact of simulation pre-training for $D(\theta)$.

We train models with the resolutions and characteristics of the Davis346 event camera [28] but demonstrate the generalization of our approach by also utilizing the Prophesee Gen3.1 VGA event camera [29] for indoor and outdoor experiments (Figure 5). These two sensors are very different, coming from two different manufacturers, with the Prophesee camera having a sensor nearly four

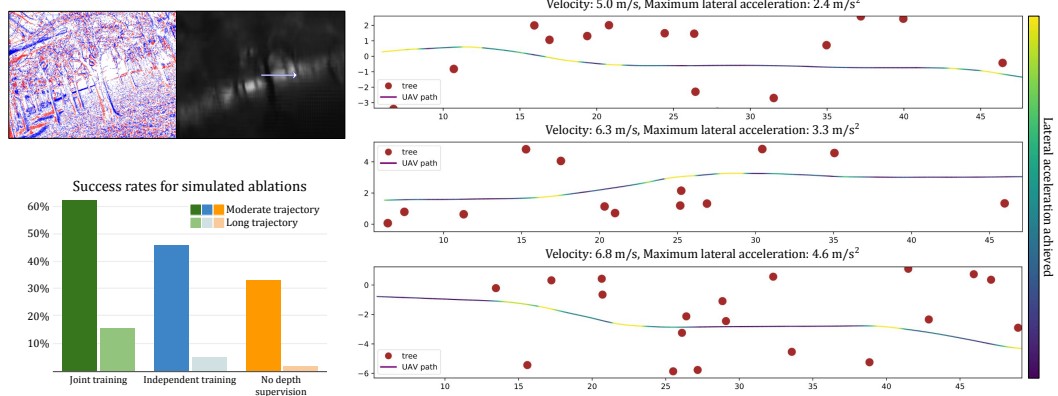

Figure 4: Simulation trials conducted in Flightmare, where approximated events (top left) are used to produce predicted depth and velocity commands. Execution success rates presented across ablations of our training method (lower left) show that jointly training the perception $D(\theta)$ and velocity $V(\phi)$ modules is beneficial. Note that the low observed simulation success rates result from the out-of-distribution event stream approximation (Equation 3) that must be resorted to during real-time execution due to the lack of a continuous-time event camera simulator.

times larger and a wider dynamic range, but also a larger latency, than the Davis. When transferring simulation models to the Prophesee camera, we similarly fine-tune with calibrated Prophesee-Realsense data (and center-crop a desired resolution of $260 \times 346$). Further details of deployment code and methods are in the supplementary.

## 4 Results

### 4.1 Simulation

We perform 100 rollouts of the trained policy in a simulated forest environment, with randomized configurations of trees unseen during training (Figure 4). However, since we deploy in simulation in real-time, we are limited to approximating the live, incoming event stream via the difference of log images. Given consecutive images $im_0$, $im_1$, the number of events at a pixel $(x, y)$ between the corresponding timestamps is as follows:

$$\hat{N}_{events_{x,y}} = \left\lfloor \left( \log(im_{1_{x,y}}) - \log(im_{0_{x,y}}) \right) / C^{\pm} \right\rfloor \tag{3}$$

This makes the input event stream out-of-distribution for models trained via Vid2E generation, so we achieve success rates around 60% with our model on a moderate-length trajectory of 10m (Figure 4, Success rates for simulated ablations, *Jointly trained*). If we increase the desired travel distance to 60m, we achieve zero collisions on 15% of trials. Other collision metrics are shown in the supplementary.

**Ablations.** The perception backbone and velocity predictor are jointly trained (see Section 3.3), such that the velocity predictor learns to interpret the depth predictions produced by $D(\theta)$. These depth images can exhibit artifacts where there are few events, and hallucinate parts of objects that are not consistent in time (see supplementary). We ablate this choice of joint training by training the perception backbone and velocity predictor independently, where the velocity predictor is given ground truth depth images. Figure 4 (lower left) shows that over 100 trials, the independently trained model achieves less than half the success rate of joint training on longer trajectories. We also train a model where depth supervision is removed entirely. This model achieves the lowest success rate, showing that depth prediction is a useful step in learning velocity commands for obstacle avoidance from event vision.

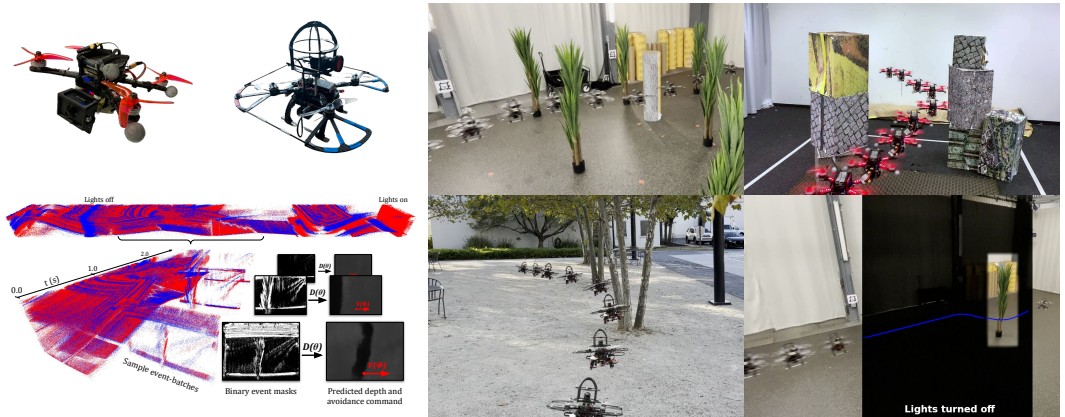

Figure 5: (top left) The two quadrotor platforms used for indoor and outdoor experiments with different event cameras. The Falcon250 [30] additionally has a VOXL board [31] for state estimation. (lower left) An event-volume (10% true density) produced by the continuous event stream from the in-the-dark trial, with select event batches, corresponding BEMs, and network predictions shown. Note the high density of negative (blue) events and positive (red) events when the lights turn off and on, respectively. (right) A variety of real experiments in indoor and outdoor conditions. Our event camera-equipped quadrotor can avoid obstacles in the dark, where traditional cameras would fail.

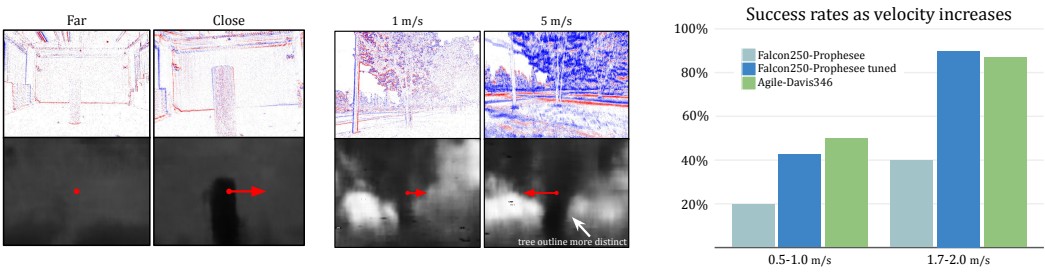

Figure 6: In the real world, events-based obstacle avoidance performs better as velocities increase, contrary to prior work using traditional vision. This is clear from the model's improved predicted depth when the obstacle is moving faster in the camera view due to proximity (left) and during faster flight outdoors (middle), and is shown in improved success rates across both platforms, even when the event camera biases for one platform is well-tuned ('Falcon250-Prophesee tuned', discussed in more detail in the supplementary). This is due to the larger volume of events seen from the obstacle of interest with increased ego-motion.

## 4.2 Hardware experiments

We conduct hardware experiments with two platforms using different event cameras (see Section 3.4, Figure 5). Results are presented for indoor and outdoor trials, where odometry is measured from motion capture and onboard VIO, respectively. As seen in Figures 6(a) and 6(b), the depth images formed by $D(\theta)$ can be amorphous, with close depth on the relevant obstacles but noisy backgrounds on textured surfaces and other objects. Furthermore, the predicted depth of the obstacle may not conform to the object's outline and it may evolve with time (see supplementary video). While certain elements of the learning framework improve this, such as the ConvLSTM layer, we find that the simulation pre-trained convolutional network $V(\phi)$ is not always able to avoid obstacles well with real-world depth predictions. Therefore, we use the open-source, obstacle avoidance pre-trained ViTLSTM (combination vision transformer-lstm) velocity prediction module [14] which contains a SegFormer-based, lightweight depth-to-velocity predictor with high generalization capabilities.

**Indoor.** For both quadrotor-event camera platforms and the corresponding environments, indoor trials were conducted in motion capture environments with tree-like obstacles that were used to fine-tune the perception backbone offline. 18 and 20 trials were conducted for the Agile-Davis346 and Falcon250-Prophesee, respectively, at varying forward speeds grouped into two bins as seen in Figure 6(c). While non-event-based obstacle avoidance methods in prior work deteriorate in performance as speeds increase, this data shows events-based avoidance notably performs better at 2m/s than 1m/s indoors. By observation, this is due to the increase in obstacle-related event volume resulting from increased ego-motion, which helps parse obstacles from background textures. This is particularly the case when facing an obstacle head-on, compared to passing by, and also at a far distance as in Figure 6(a) (left). At up to 1m/s forward speed, not enough events are seen on the object, resulting in a poor depth estimate and often a collision. We also conduct trials in fields of multiple obstacle types as well as in the dark (Figure 5), where traditional depth cameras fail to identify obstacles.

**Outdoor.** Outdoor tests were conducted with the Falcon250-Prophesee platform on real trees in an urban environment, where the background scene includes stopped and moving cars, sidewalks, buildings, and other trees. At the times of experimentation, wind speed was 8-16kph with gusts of 24-32kph. After training the model on these trees, we deployed it with desired speeds of 1-5m/s and observed a 11/13 (85%) success rate, where the only two failures (collisions) occurred when the quadrotor tried to avoid right with low battery but was pushed left by wind gusts. Notably, compared to indoor testing with the Falcon250-Prophesee, we observed more consistent depth prediction on the outdoor trees, resulting in better avoidance capacity. This may be due to the existence of certain high-volume event emitters indoors (motion capture and industrial lighting), as well as the artificial and patterned background textures in the test space that may obscure the event patterns on an obstacle until very close. Furthermore, similarly to the indoor results, as forward speed increased from 1m/s to 5m/s, the quadrotor exhibited *better* avoidance ability in terms of magnitude and earliness of the avoidance maneuver, contrary to prior work in high-speed vision-based avoidance where performance tends to decrease with increasing velocity [12].

## 5   Conclusion

We present simulation and real-world results demonstrating that we can avoid static obstacles with only an event camera onboard a moving quadrotor. We rely on a frame-based representation of the event stream, which, as we do not have a continuous-time events simulator, allows simulation pre-training of a policy that can then be fine-tuned with real-world perception data. Data augmentation applied to the binary event mask representation helped overcome the large events sim-to-real gap and even the real-to-real gap (stemming from varying lighting, sensor noise, and background textures). Simulation results support that supervising depth as an intermediate step in the network improves success rates on both moderate and long trajectory lengths.

Our real-world experiments suggest that unlike traditional perception-based obstacle avoidance, event-based avoidance performs better at high speeds (5m/s) than at low speeds (1m/s), as more events are seen on obstacles which improves depth prediction quality and thereby the avoidance action. Also notably, outdoor tests had a higher success rate than indoor tests done on the same platform, likely due to the artificial event sources and patterned background textures indoors that can obscure textured obstacles.

**Limitations.** This work focuses on the use of event cameras for obstacle avoidance, and demonstrates surprising benefits of this perception modality, including flying in the dark. However, we do not directly compare the benefit of an event camera's low sensor latency against that of a traditional vision sensor, in part due to the processing latency of model inference, which is currently magnitudes larger than the event camera's sensor latency. Please refer to the supplementary for a more detailed discussion of these latencies. Finally, though collecting real-world perception data is inexpensive compared to collecting expert pilot data, it is nevertheless required to fine-tune our model with calibrated events-depth data for novel scenes and objects.

## Acknowledgments

This work was partly supported by the European Research Council (ERC) under grant agreement No. 864042 (AGILEFLIGHT), and by NSF award ECCS-2231349, SLES-2331880, and CAREER2045834. From the Robotics and Perception Group, we thank Alex Barden, who engineered the Agile-Davis platform, Ángel Romero, who assisted in initial experimentation and debugging, Nico Messikomer, who provided input in events-based learning pipelines, Daniel Gehrig and Mathias Gehrig, who advised on general directions, and Yunlong Song, who helped on simulation techniques. From Kumar Lab, we thank Alex Zhou, who helped develop hardware for the Falcon250 platform and attachments, Yuwei Wu and Fernando Cladera, who provided input and hands-on support when debugging experimental issues, and Alice Li and Xu Liu, who advised in outdoor experimentation.

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

# Supplementary material for: Monocular Event-Based Vision for Obstacle Avoidance with a Quadrotor

## S1   Simulation details

### S1.1   Simulation environment

We utilize the Flightmare simulator [1] to collect photorealistic perception data while running a privileged expert [2] through a forest-like environment. This simulator package runs the Unity rendering engine, and to generate randomized forest-like environments for training we place and orient 100 trees uniform-randomly across a 40m × 50m area. A grayscale image of the scene, the corresponding events batch (computed offline via Vid2E [3]), and the depth image are shown in Figure S1.

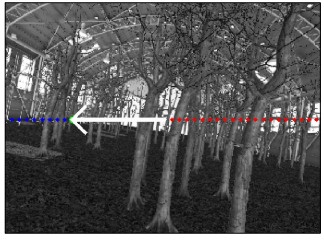
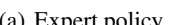

(a) Expert policy.

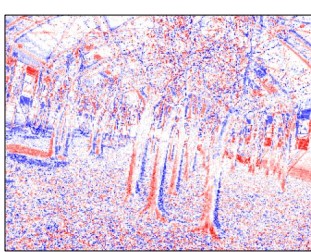

(b) Corresponding event batch.

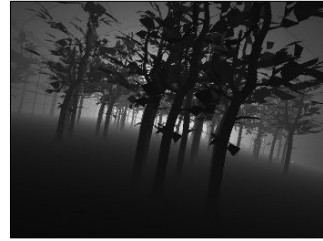

(c) Corresponding depth image.

Figure S1: One timestamp of an expert policy trajectory in simulation. (1(a)) The expert chooses the closest collision-free waypoint (free waypoints represented as blue dots, opposed to collision waypoints in red) for planning. (1(b)) The event batch is computed offline via Vid2E, and used to train $D(\theta)$ with the (1(c)) depth image as supervision.

### S1.2   Privileged expert

Our dataset for training both the perception backbone and velocity predictor consists of 100 trajectories of the privileged expert traveling through a forest-like environment. For each trajectory, the expert's forward speed was uniform-randomly selected between 3-7m/s. The privileged expert has access to obstacles within a 10m radius of its current position and calculates a collision-free waypoint relative to inflated obstacles to issue velocity commands towards in a receding horizon fashion. Query points are along a horizon-aligned line 10m ahead of the drone, and at 5Hz the collision-free waypoint closest to the center of the line (corresponding to forward flight) is chosen, then multiplied by a constant gain to calculate velocity commands. Figure S1 shows the corresponding data; 1(a) depicts a white arrow with the commanded velocity.

### S1.3   Additional simulation collision metrics

We present additional collision metrics from simulation rollouts for the proposed model and ablations described in Section 4.1 (jointly-trained, independently-trained, and no-depth-supervision models) in Figure S2. Statistics are calculated from 50 trials per reported metric, across forward velocities ranging uniformly between 3-7m/s. As we allow one or two collisions in the success rate calculation (typically "success rate" refers only to zero collisions, as in the main text) we observe a

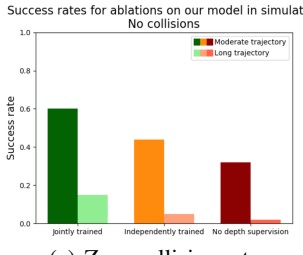
(a) Zero collision rates.

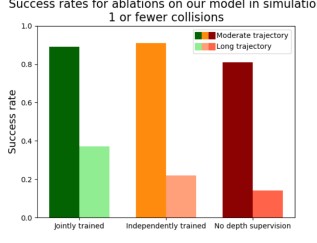
(b) One or fewer collision rates.

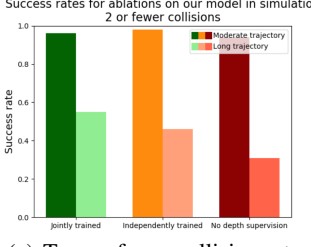
(c) Two or fewer collision rates.

Figure S2: Rates for moderate and long trajectories in simulation of zero collisions (typically referred to just as "success rates") and one and two collisions. On the long trajectory there is consistently better performance with the jointly-trained (proposed) model.

leveling out of performance among all models on the moderate trajectory, but performance *remains consistently better* on the long trajectory with the jointly trained model.

## S2  Event camera hardware and calibration with a depth camera

Two quadrotor-event camera platforms were used, one with a Davis346 (resolution $260 \times 346$) and one with a Prophesee Gen3.1 VGA (resolution $480 \times 640$) (Figures 5). As mentioned in the text (Section 3.4), these two sensors have very different sensor and event-pixel characteristics; furthermore, the biases were left as the default values for most experiments. To enable the simulation pre-trained policy to extend to a real-world Davis346 and further to the Prophesee camera, calibrated and time-synchronized event batches and depth images were gathered to fine-tune models with real data gathered from a hand-held device with both sensors (Figure S3). ROS was used to run both sensors in parallel while recording message data that includes ROS timestamps and corresponding events messages for either event camera (iniVation AG ROS driver [4] was used for running the Davis346, and Prophesee AI ROS driver [5] or a 3rd-party Metavision ROS driver [6] was used for running the Prophesee) as well as ROS timestamps and corresponding depth and "infrared1" messages from the D435 depth camera. Since the "infrared1" camera is pre-aligned with the depth images, we use these images to run multi-camera calibration. E2Calib [7] was used to generate images from either event camera at timestamps specified by the depth images (for both types, rosbag data needed to be converted to DVS EventArray message type [8, 9, 10]). With the images from E2Calib and "infrared1", we use Kalibr [11] to perform multi-camera calibration.

As all perception models are sized according to the Davis346 resolution, for the Prophesee camera we center-crop a $260 \times 346$ sized event batch for input to the perception backbone $D(\theta)$. Note that while the simulated camera and corresponding Vid2E events output have different intrinsics than the Davis346 and Prophesee cameras, no alignment is performed during the sim-to-real transfer, and we expect the fine-tuning training of the perception backbone to incorporate changes in lens geometry and distortion.

## S3  Computational details for simulation and hardware

We provide computational details for the simulation computer and the Falcon250-Prophesee hardware platform in Table S1. For an indirect comparison, [12] utilizes Jetson TX2 onboard compute for depth image-to-trajectory planning, where inference time is 38.9ms. In contrast, we use event streams directly as the perception input, predict a depth image intermediately, and predict velocity commands in an end-to-end event-based perception-to-control framework, with total onboard inference time of 73ms. Note that further optimization of the computational complexity and processing latency may be possible, as this was not the focus of this work. Equation 11 of [12] supplementary material specifies how reduction in sensing time $t_s$, as may be possible with event cameras over traditional cameras, may increase the theoretical maximum velocity achievable while avoiding a

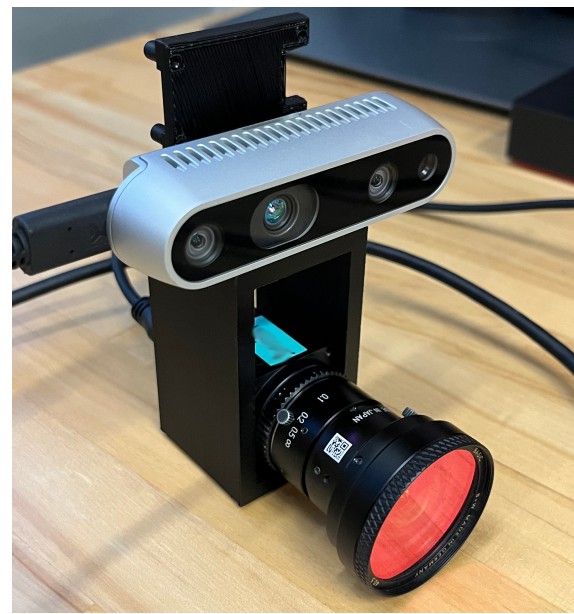
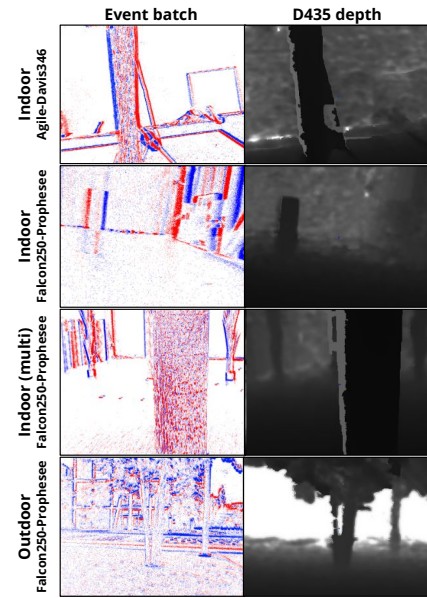

(a) Handheld rig for real-world data collection.   (b) Sample event batch-depth training pairs.

Figure S3: Example of rigidly-attached event camera (Prophesee Gen3.1 VGA) and depth camera (Intel Realsense D435) used to gather events batches and depth images for approximate post-processing calibration and time synchronization, for use in fine-tuning perception backbone from Figure 3. Note that an IR filter is used on the event camera lens so that we may enable the infrared emitter on the D435 for better depth image precision.

|  | Simulation | | Falcon250-Prophesee | |
|---|---|---|---|---|
|  | Num. parameters | Inference time [ms] | Num. parameters | Inference time [ms] |
| $D(\theta)$ | 9,856,673 | 255 | 9,856,673 | 67 |
| $V(\phi)$ | CNN | | ViTLSTM | |
|  | 923,321 | 3 | 3,563,663 | 6 |
| **Totals** | 10,779,994 | 258 | 13,420,336 | 73 |

Table S1: Model size and inference time for simulation (CPU-only Intel Core i7-10710U, 16GB RAM) and for hardware experiments with the Falcon250-Prophesee platform (CPU-only Intel Core i7-10710U, 32GB RAM). We provide inference times for the perception backbone $D(\theta)$ and the velocity prediction $V(\phi)$. Note that for simulation $V(\phi)$ consists of a CNN, whereas for real-world experiments $V(\phi)$ consists of a ViTLSTM model [2].

given cylindrical (i.e., tree-like) object. While the current work batches events by 33ms windows, this batch can be updated with incoming events in real-time, at frequencies only upper-bounded by the pre-processing required of events. This effectively utilizes the low sensing latency $t_s$ of an event camera. In this work, pre-processing requires 650$\mu$s, or 2.250ms with an optional event-frame alignment step to better match training data (based on event camera and depth camera calibration). Given these pre-processing latencies, and the formulation given in [12] with the same drone parameters, we would be able to increase the theoretical maximum avoidance flight velocity from 13.5m/s to 15.83m/s or 15.76m/s, respectively. Considering the total sensing and processing latency $t_s + t_p$, [12] cites $66 + 10.3 = 76.3$ms, and this work, as specified above, would have a worst-case latency of $2.25 + 73 = 75.25$ms.

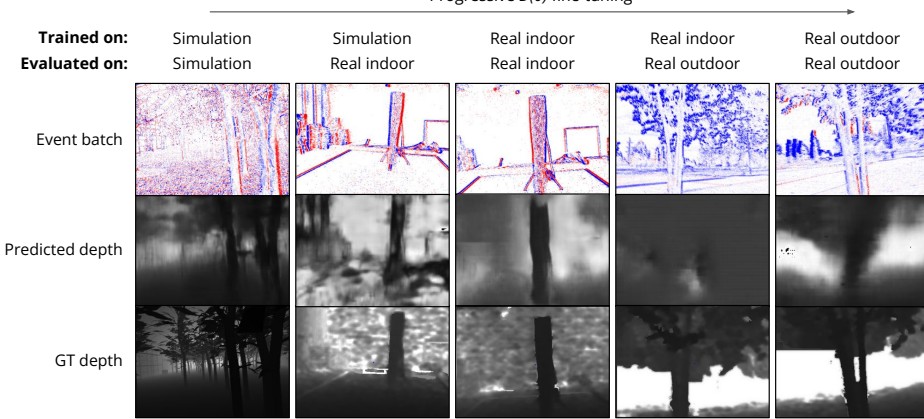

Figure S4: Progressive evaluations on more challenging environments, ranging from simulation to real indoor to real outdoor. A simulation pre-trained model is evaluated in simulation, then on a real indoor environment before and after fine-tuning, and so on.

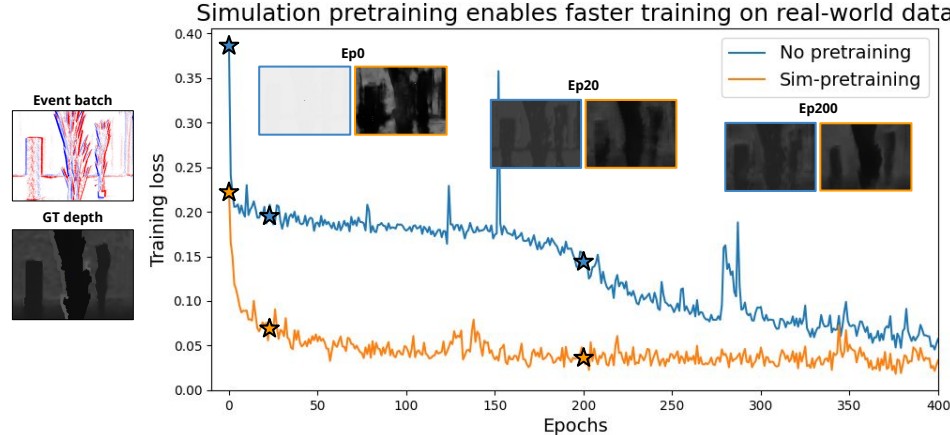

Figure S5: Simulation pre-trained (orange) events-to-depth models train faster on real data than non-pre-trained models (blue). Figure inserts contain evaluation examples of the predicted depth at epochs 0, 20, and 200 of training both models on real indoor data. Input event batch and ground truth depth are on the left of the plot.

## S4 Pre-training and fine-tuning characteristics of $D(\theta)$

### S4.1 Generalization and fine-tuning

In Figure S4 we provide generalization and fine-tuning examples of the perception backbone $D(\theta)$ pre-trained in simulation, then applied and fine-tuned to progressively more difficult real-world scenes. The simulation-trained model's evaluation on real indoor data produces a recognizable depth image, which is fine-tuned to achieve appropriate relative scaling and in-filling.

### S4.2 Pre-training speeds up fine-tuning on real scenes

Training loss curves during training of $D(\theta)$ on real indoor event camera data for a simulation pre-trained model and non-pre-trained model are shown in Figure S5. The pre-trained model produces a recognizable depth image immediately and fine-tunes to the real data in less than 20 epochs, bar some improvements in consistency, whereas by epoch 200 the non-pre-trained model has yet to learn the relative scaling and edges of the obstacles.

## S5 State estimation for outdoor flight

As described in the main text, the Falcon250-Prophesee platform uses the VOXL board [13] for onboard state estimation. This board contains an independent computer, connected to the primary computer via an Ethernet cable. This visual-inertial odometry (VIO) module provides six degrees-of-freedom robot poses at 30Hz. We employed an Unscented Kalman Filter (UKF) that takes high-frequency IMU data in combination with these pose estimates to provide odometry at 150Hz. Low-level attitude and thrust controllers are run on PX4 open-source [14] control stack using the Pixhawk4 flight controller, with desired values provided by custom controllers running on the primary onboard computer.

## S6 Tuning event camera biases for increased performance

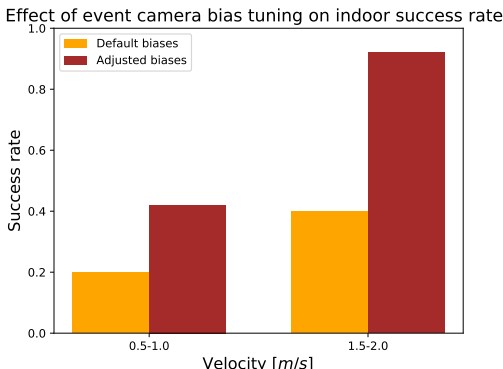

Figure S6: Success rates for the Falcon250-Prophesee platform avoiding an obstacle indoors, with and without adjusted biases meant to reduce background textural and random photon arrival noise. This tuning results in a double of the success rate while maintaining the observed upwards trend in success rate as forward speed increases.

| Bias name | Default value | Adjusted value |
|---|---|---|
| bias_diff | 299 | 299 |
| bias_diff_on | 384 | 400 |
| bias_diff_off | 222 | 200 |
| bias_fo | 1477 | 1590 |
| bias_hpf | 1499 | 1499 |
| bias_refr | 1500 | 1400 |

Table S2: Default and adjusted values for Prophesee Gen3.1 event camera biases used for additional indoor testing with the Falcon250-Prophesee platform. Changes include widening the contrast threshold gap (bias_diff_on, bias_diff_off), narrowing the sensor bandwidth (bias_fo), and reducing refractory period (bias_refr).

Indoor hardware results, as well as some outdoor results, in the main text are deliberately achieved *without tuning of event camera biases* for either event camera used, as we aim to show trends that appear naturally from our simulation-to-real, few-shot transfer to multiple hardware platforms. For example, we observe increasing success rates as flight speed increases, and also a higher success rate as we transition to an outdoor setting. However, as mentioned in Section 3.4, the Prophesee camera has a four-times-larger sensor than the Davis346, and thus, when paired with no additional bias tuning, produces significantly higher "noise" levels due to both background textures and random photon arrival (see Figure 6(a), top row). Through observation, this results in a lower success rate than that achieved by the Agile-Davis346 platform.

This background level of events is particularly strong in the chosen indoor testing environment, where motion capture, industrial lighting, and surrounding patterned, artificial textures create large volumes of events (discussion in Section 4.2). Here, we present analogous results from the same indoor conditions but with slightly tuned event camera biases (Table S2). Figure S6 compares success rates with (24 trials) and without (20 trials) tuned biases for the Falcon250-Prophesee platform. We observe a doubling of the success rate. Other tuning may improve performance further, including training with varied time window batching of events under these improved biases, so the model may predict obstacle depth accurately from the fewer ego-motion-induced events.

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
