# OpenReview forum: "Monocular Event-Based Vision for Obstacle Avoidance with a Quadrotor"
_robot-learning.org/CoRL/2024/Conference — CoRL 2024_

### Official Review · Reviewer_tqUw · 2024-07-20
**Review for Paper 539**

**Originality:** 4
**Technical Quality:** 4
**Clarity Of Presentation:** 4
**Potential Impact:** 3
**Recommendation:** 3
**Confidence:** 4

**Review:**

**Strengths**
- This method tackles the difficult problem of learning a obstacle-avoidance quadrotor controller for event-based cameras despite having direct access to sufficient event training data
- The authors present a unique training pipeline to generate event data from high fidelity image data collected in simulation, thus sidestepping the problem of not having access to a simulator capable of generating event data for their specific problem.
- The method is capable of few-shot transfer to real hardware

**Weaknesses**
- The method performs better at higher velocities (>1 m/s) and worse at lower velocities (<1 m/s). The max velocity reported is 5m/s. Note that prior works in vision and non-vision based quadrotor control operate at a similar 1-5 m/s range and do not suffer from the issue with event-based cameras at low velocities. I understand that both CMOS camera and event camera based visual control methods have tradeoffs. However, in the case of CMOS cameras, one can simply upper-bound the speed of the quadrotor to avoid performance degradation. Unfortunately, I don’t see how this approach would be useful in any scenario if the quadrotor can only fly “fast” to obtain sufficient event data just to avoid a collision with a static obstacle. This seems like a significant downside of this approach.

- Figure 3b shows us an example scenario the authors evaluate their method on. The obstacle density seems quite sparse, so much so that the quadrotors rarely need to deviate from their nearly linear flight path. Yet the highest reported success rate is 60%, which is quite odd. I understand that the distribution shift caused by using Vid2E is responsible, but if there is no workaround to this issue, them I’m afraid compared to prior methods that operate at similar speeds with higher success in similar if not more obstacle dense environments, these results to me seem like a step backwards.

**Quality Of The Limitations Section:**

3

**Questions For Rebuttal:**

1. When the velocity predictor is trained independently on GT data, the performance is worse. This implies that there must be some significant distribution shift between GT and generated depth maps (otherwise the performance drop from joint —> independently trained wouldn’t be so severe) indicating either 1. A lack of sufficient training data / training time for the depth network, 2. Suboptimal training procedure or 3. Both. The authors suggest one issue is temporal consistency. Did the authors try using a stack of consecutive depth maps to predict the depth map at the next timestep? Or maybe accumulating depth information through time using an RNN? Seems like this issue significantly hinders results yet seems solvable with current methods.

**Robotics Focus:**

3

**Summary Of Paper:**

This method proposes using event-cameras for learning quadrotor control and obstacle avoidance policies from visual observations.

**Summary Of Recommendation:**

The authors present a novel method of learning static obstacle avoidance for quadrotors using event cameras by leveraging depth estimation from events as an intermediate step. The novelty of the approach and qualitative and quantitative results at high velocities merit acceptance. However, the lack of sufficiently challenging scenarios and the poor performance at lower velocities, which may also be due in part to limitations of the method, hold it back from being a strong paper.

---

### Official Review · Reviewer_gGzy · 2024-07-21

**Originality:** 3
**Technical Quality:** 3
**Clarity Of Presentation:** 4
**Potential Impact:** 3
**Recommendation:** 3
**Confidence:** 4

**Review:**

This work provides an interesting proof of concept for addressing what has hitherto been regarded as a weakness of event cameras, undoubtedly pushing the boundaries of what is possible with event-based vision and indeed reactive control more generally. I am pleased to recommend its acceptance, however I believe that there are two areas of improvement which, if addressed, would help readers better appreciate the impact of this work:

(1) Comparison with alternatives: Despite the impressive first-of-their-kind results, for agile flight practitioners interested in deploying a quadrotor collision avoidance solution it is not clear whether this technology is something immediately worthy of refinement and deployment, or whether the purpose of this work is simply to demonstrate the value of certain individual design decisions within a proof-of-concept framework. It would be helpful to provide a more in depth comparison placing this work in context with alternative sensing solutions currently used in practice (e.g., traditional CMOS cameras including global shutter cameras, depth cameras, on-board LiDAR, etc.). Even qualitatively it would be informative to understand what levels of motion blur are actually observed in real-world flight at these speeds, as eliminating motion blur seems to be the primary utility provided by the event camera in this work. Indeed, a major limitation that the authors note is that their framework, by frame batching and running heavy neural network computation on these frames, sacrifices one of the perceived advantages of event cameras, namely near-zero sensor latency. What would the ideal application setting (e.g., what environments, what speeds) for this technology be, where it might have the potential to outperform alternative solutions?

(2) More discussion on generalization performance: The last sentence of this work states "our model is fine-tuned on new scenes and objects, but does not generalize to new obstacles" -- what is the distinction drawn here between "objects" and "obstacles"? How does the depth prediction look in sim/real for scenes other than column/tree-like objects? Generalization is the key challenge in transfer learning, and this paper is quite sparse in details on (i) how the sim training environments were generated, (ii) diversity of expert policy behavior, (iii) the coverage of the real fine-tuning data. Related to point (1), if the Realsense D435 depth camera is trusted as a source of ground truth depth, is it the case that (i) motion blur is not actually an issue at the speeds the quadrotors are flying at (and a different method needing no ground truth depth data for fine tuning would be needed in higher-speed regimes) or (ii) the fine-tuning data was gathered at lower speed compared to the full-speed experiments (in which case generalization to higher speeds is transferred from training in simulation)?

**Quality Of The Limitations Section:**

2

**Questions For Rebuttal:**

From the comments/questions above, please prioritize:
- What would the ideal application setting (e.g., what environments, what speeds) for this technology be, where it might have the potential to outperform alternative solutions?
- How do the "real datasets used for fine-tuning consist[ing] of 10-20 shorter trajectories" compare to the deployment conditions (e.g., speed) of the real-world experiments, and how was the real depth data validated?

**Robotics Focus:**

4

**Summary Of Paper:**

In this paper the authors propose a sim-to-real transfer learning framework for vision-based autonomous flight with event cameras, and use it to demonstrate, for the first time, static obstacle avoidance with such a sensing setup. Methodologically, the authors demonstrate the value of depth prediction as an auxiliary pre-training task for learning a lateral velocity collision avoidance policy with a privileged teacher-student approach; they also propose a frame-batching approach to processing the event stream that allows the application of established neural network architectures and data augmentation techniques from the computer vision literature. Experiments demonstrate the efficacy of the sim-to-real transfer and overall policy learning framework, for indoor and outdoor flight at high speeds.

**Summary Of Recommendation:**

I recommend this paper's acceptance at CoRL 2024 as it brings to the robot learning community an interesting sensorimotor policy learning problem (versatile events-to-control flight) and demonstrates how recent ideas in teacher-student transfer learning may be adapted to provide a solution. However, as noted above, I believe the paper would be improved if the practical utility of this work could be more clearly established.

---

### Official Review · Reviewer_n5tQ · 2024-07-21
**Good experiments and demonstrations, but slow for event-based cameras and limited obstacle diversity**

**Originality:** 3
**Technical Quality:** 4
**Clarity Of Presentation:** 4
**Potential Impact:** 3
**Recommendation:** 3
**Confidence:** 3

**Review:**

Quality: The paper is of good quality, and presents a well-structured approach for a challenging problem in robotics. Thorough experiments in both simulation and real-world settings were conducted, using multiple platforms. The methodology is sound, with clear justifications for design choices and ablation studies to support their approach.

Clarity: The paper is generally written well and organized well too. They clearly state their objectives, methodology, and results. The figures and diagrams help illustrate the key concepts and results.

Originality: The work presents several novel contributions. The authors also mention that this is the first work to demonstrate events-only static-obstacle avoidance for a quadrotor with significant ego-motion.

Significance: This research could have significant implications, especially in scenarios requiring high-speed navigation in cluttered environments. The finding that event-based dodging improves at higher speeds is particularly noteworthy and could influence future research directions.


Strengths:
- First paper demonstrating static obstacle avoidance for a quadrotor using an event-based camera.
- Successful sim-to-real transfer using a novel approach that leverages depth prediction.
- Demonstration of high-speed (5m/s) obstacle avoidance using only onboard computation.
- Experimentation across multiple platforms and environments (indoor and outdoor).
- Counter-intuitive finding that performance improves at higher speeds, contrary to traditional vision-based approaches.

Weaknesses:
- Limited exploitation of event camera advantages (low latency, high dynamic range) due to computational constraints. The computational complexity limits the maximum achievable speed, which undermines the initial motivation.
- Limited generalizability - requires fine-tuning for new environments and obstacles. In a practical scenario, a drone using this system couldn't simply be sent into a new area without first collecting data and retraining the model, which is time-consuming and resource-intensive. For fine-tuning, the authors collected aligned event batch and depth image pairs using an event camera rigidly mounted with a depth camera. This process requires specific equipment and setup, which may not always be feasible in all environments where the drone might need to operate.
- Due to the above, it does not fully realize the goal of high-speed, agile flight through completely unknown environments, as initially motivated.
- Focuses primarily on “tree-like” obstacles, with limited exploration of various other obstacle types.

**Quality Of The Limitations Section:**

3

**Questions For Rebuttal:**

Please provide more details on the limitations of generalizability. How much fine-tuning is required for new environments? How different is the data that the model is fine-tuned on from the data that it sees at test-time in the new environment? Can you quantify the performance drop when the system encounters entirely new obstacles or environments without fine-tuning?

The paper mentions that computational requirements limit the maximum achievable speed. Can you provide more details on the computational resources used?

Given that the low latency of event cameras isn't fully utilized due to computational constraints, how does your approach compare to traditional vision-based methods in terms of overall system latency and performance? At the speeds that your system is capable of operating at, do they outperform traditional vision-based methods? Can you add experiments in simulation that compare the performance of the two?

The focus is primarily on tree-like obstacles. Can you provide data on performance with a wider variety of obstacle types and shapes? This would help elucidate how much the approach might have overfit to this particular type of obstacle.

**Robotics Focus:**

4

**Summary Of Paper:**

This paper introduces the first events-only static-obstacle avoidance method for quadrotors using a monocular event camera. It leverages depth prediction to guide the network in extracting representations. The approach involves pre-training a reactive obstacle avoidance policy in simulation and fine-tuning with limited real-world data. Key contributions include demonstrating the first purely events-driven obstacle avoidance for a robot with significant ego-motion, successful sim-to-real transfer for tree dodging, and achieving purely onboard computation at 5 m/s. The method excels at higher speeds and shows better performance in outdoor settings. Event cameras enable vision-based agile flight at faster speeds due to their high dynamic range and motion blur elimination.

**Summary Of Recommendation:**

The paper presents a novel approach to static obstacle avoidance for quadrotors using event-based cameras, demonstrating success in both simulated and real-world environments. The work is of good quality, clearly written, and offers original contributions to the field. Its significance lies in showcasing high-speed navigation in cluttered environments and the counter-intuitive finding that performance improves at higher speeds. However, there are limitations in generalizability and full exploitation of event camera advantages due to computational constraints. The research has strengths in being the first to demonstrate this capability for quadrotors and successful sim-to-real transfer, but weaknesses in limited obstacle types and the need for fine-tuning in new environments.

---

### Author Rebuttal · Authors · 2024-08-11

5ms_outdoor: videos of paper's 5m/s outdoor trial, showing eventstream with two different visualizations.

dynamic-obstacle: new experiment showing the drone dodging an approaching obstacle.

low-light: new experiment showing dodging under near-total darkness where CMOS cameras fail (further supplementary figures yet to come).

multi-obstacle: new experiment showing the dodging of more types of obstacles within a single trial.

bias-tuning: a video of 6 of 24 new trials done with tuned biases for the Falcon250-Prophesee platform.

supplementary-updated_8-13.pdf: updated supplementary addressing reviewer concerns. On 8-11 added Section S6 Simulation trajectory samples; on 8-13 added S7 Tuning event camera biases for increased performance.

While we have encoded videos with accessible encodings, please let us know if anything is not playable or needs clarification.

---

### Decision · Program_Chairs · 2024-09-04

**Decision:**

Accept

**Comment:**

The submission proposes using event cameras to learn drone control for evading static obstacles.
This submission proposes using event-based cameras for quadrotor obstacle avoidance. The authors successfully demonstrate a novel approach for transferring a simulation-trained model to real-world conditions, achieving high-speed obstacle avoidance with onboard computation. Their experimentation across multiple platforms and environments showcases the method's robustness. Additionally, the counterintuitive finding of improved performance at higher speeds is useful.
The paper lacks a comprehensive comparison with alternative sensing solutions, making it difficult to assess the method's impact. The current approach is limited by computational constraints, hindering the full exploitation of event camera advantages like low latency and high dynamic range. The method's reliance on fine-tuning for new environments limits its generalizability. Furthermore, the focus on tree-like obstacles restricts the scope of the research.
The paper has improved during the rebuttal via detailed communication between reviewers and authors. Based on the above the paper is recommended for acceptance. Please follow up on the remaining open points for a potential camera ready version. With new results on dynamic obstacles, it will be worth reconsidering the positioning of this work slightly.